# Effect of 580 °C (20 h) Heat Treatment on Mechanical Properties of 25Cr2NiMo1V Rotor-Welded Joints of Oscillating Arc (MAG) Narrow Gap Thick Steel

**DOI:** 10.3390/ma14164498

**Published:** 2021-08-11

**Authors:** Xiaoyan Qian, Xin Ye, Xiaoqi Hou, Fuxin Wang, Shaowei Li, Zhishui Yu, Shanglei Yang, Chen Huang, Jinpeng Cui, Chunxiang Zhu

**Affiliations:** 1School of Materials Engineering, Shanghai University of Engineering Science, Shanghai 201620, China; qianxiaoyan2019@163.com (X.Q.); houxiaoqi2019@163.com (X.H.); wangfuxinhit@sues.edu.cn (F.W.); henhen_18@163.com (S.L.); yu_zhishui@163.com (Z.Y.); yslei@sues.edu.cn (S.Y.); hch@sues.edu.cn (C.H.); 2Shanghai Collaborative Innovation Center for Advanced Laser Manufacturing Technology, Shanghai 201600, China; 3State Key Laboratory of Metal Material for Marine Equipment and Application, Anshan 114001, China; 4Shanghai Turbine Factory, Shanghai Electric Power Station Equipment Co., Ltd., Shanghai 201100, China; cuijip@shanghai-electric.com (J.C.); zhuchx@shanghai-electric.com (C.Z.)

**Keywords:** swing narrow gap MAG welding, microstructure, impact, tensile

## Abstract

The thick plate narrow gap welding of 25Cr2NiMo1V rotor steel is achieved by metal active gas arc welding, in which the weld gap was 18.04–19.9 mm. After welding, the weldment was heat treated at 580 °C (20 h). The impact and tensile properties in the as-welded and heat-treated were studied. The results show that after heat treatment, the coarse carbides in the center of the weld were transformed into fine granular carbides distributed along the grain boundaries, and the quantity of carbide precipitates in the weld near the fusion line was reduced. The tensile fracture mode changed from a ductile fracture to a combination of brittle and ductile fractures, and the tensile strength of the weld metal changed from 605 MPa to 543 MPa. After heat-treated, the radiation zone of the weld center changed from a brittle fracture to a combination of brittle and ductile fractures, and the impact energy changed from 141 J to 183 J; the characteristics of the brittle fracture in the radial zone of the fusion line were more obvious, and the impact energy changed from 113 J to 95 J. Therefore, after heat treatment, the toughness of the welded metal was improved, without reducing the strength and hardness of the welded metal to a large extent.

## 1. Introduction

Due to the rapid development of society, the demand for electricity is steadily increasing, resulting in a large quantity of emissions of carbon dioxide and other gases, and worsening environmental pollution. Therefore, power generation technology needs to be improved to increase energy utilization and reduce environmental pollution. The development of ultra-supercritical (USC) power generation technology meets the requirements of high efficiency and low pollution; thus, this technology should be developed at a large scale [1,2,3,4].

The steam turbine rotor is the core component of the generator set [5]. Improving its performance is an important requirement for enabling supercritical power generation technology. The rotor is subjected to high temperature and complex stress, and needs to be repaired under different temperature environments. Thus, it is highly important to achieve high levels of use and material performance [6]. Hu et al. [7] studied the distribution of residual stress after welding of 25Cr2Ni2MoV rotor steel, and methods to reduce residual stress through experiments and numerical simulations. Chu et al. [8] studied the welded joints of NiCrMoV steel and found that the small grains in WM can withstand the crack driving force, reduce stress concentration, and increase the resistance to transgranular crack propagation. Guo et al. [9] studied the effect of the initial crack length of 9Cr/CrMoV dissimilar welded joints on the crack propagation resistance, and found that the longer the initial crack length, the lower the crack propagation resistance. Wang et al. [5] used oil coating (BL) to weld two supercritical steam turbine rotor steels, and found that the fatigue life of the weld metal (WM) and base metal (BMs) was lower, the weld zone showed cyclic softening characteristics, and weld fatigue cracks expanded mainly due to coarse carbides. Zhu et al. [10] studied the tensile and impact behaviors of a newly developed rotor steel and 26NiCrMoV145 dissimilar welds at different temperatures below 350 °C, and observed that the dissimilar welds were uneven in structure and microhardness along the line. Wang et al. [11] studied post-welding heat treatment of 10%Cr rotor steel and found that the impact toughness was significantly improved after heat treatment. Multilayer or multipass welding can improve the weld structure, increase plasticity, and improve product quality [12,13], so is often used for thick plate welding [14]. In addition, to further increase the productivity of rotor steel welding and improve its mechanical properties, arc narrow gap welding is widely used. Narrow gap arc welding uses fewer consumables and reduces deformation. It has been widely used in medium and thick steel plates. It is an effective method for welding rotor steel [15,16].

For the welding of rotor steel, creep performance and fatigue resistance are important. However, appropriate impact toughness and tensile properties at room temperature are also important to avoid problems during testing and startup/shutdown [17].

In this study, swing narrow gap MAG welding was performed on 50 mm thick plate 25Cr2NiMo1V rotor steel, and the pattern was subjected to post-weld heat-treatment at 580 °C (20 h). Pendulum impact tests and tensile tests were carried out on the as-welded metal and after the heat treatment, and the fracture morphology and failure mode of the impact specimens and tensile patterns were analyzed. In this paper, the relationship between the toughness and microstructure are also discussed to provide theoretical support for follow-up research.

## 2. Materials and Methods

The experiment used a 25Cr2NiMo1V rotor steel thick plate, the dimension of the butt joint test plate was 400 × 120 × 50 mm^3^, the upper assembly gap was 19.9 mm, the lower assembly gap was 18.04 mm, and the groove was 2.3°. The welding wire used was JM-56 (Tianjin Sea welding Technology Co., LTD), which has a diameter of 1.2 mm. The alloy composition of the base metal (25Cr2NiMo1V) and welding wire are shown in Table 1.

A narrow gap welding swing arc welding torch was used in the experiment. The swing welding system is shown in Figure 1. The shielding gas was a mixture of CO_2_ (20%) and Ar (80%) gases, and the gas flow rate was 20 L/min. The welding voltage was 30 V, the welding speed was 20 cm/min, the wire feeding speed was 10 m/min, and the interlayer temperature was 80 °C. A total of 15 layers were welded, and the average thickness of each layer was 3.3 mm.

After welding, the same test plate was divided into two parts: one part remained in the as-welded state, and the other was subjected to post-weld heat-treatment using a heat-treatment process of 580 °C (20 h) [11]. Metallographic specimens, tensile specimens, and impact specimens were prepared using the wire-cut electrical discharge method. The metallographic sample was corroded in 4% nitric acid alcohol solution for 12 s. The microhardness was obtained by applying a test load of 200 g (1.961 N) on a Vickers hardness tester for a duration of 10 s. A Hitachi S3400 scanning electron microscope and other techniques were used to analyze the distribution of carbides, fracture morphology, and fracture location. Take 6 tensile samples at the same position of the as-welded and heat-treated weldments, and 20 impact samples were taken from the center of weld and the center of fusion line. The location and size of the samples are shown in Figure 2.

## 3. Results and Discussion

### 3.1. Tensile and Impact Testing

To study the mechanical properties of the welded metal after heat treatment, the tensile and impact properties were measured at room temperature to evaluate the tensile strength and toughness of the welded metal.

Figure 3a shows test positions of the upper, middle, and lower parts of the metal. To study the effect of heat treatment on the tensile properties, the tensile properties of the upper, middle, and lower parts of the weld were individually tested. To study the impact of heat treatment on the impact properties of the weld center and the fusion line, the upper, middle, and lower parts of the weld were individually subjected to an impact performance test. In the experiment, we extracted the impact samples at the center of the weld seam and the fusion line, and the sizes of the two samples were the same. Of these, the sample at the fusion line was extracted by aligning the fusion line with the v-shaped notch, and the sample at the welding seam was extracted by aligning the center of the weld with the v-shaped notch. Figure 3b shows the changes in tensile strength of the welded and heat-treated joints. The results showed that the tensile strength of the upper, middle, and lower parts of the as-welded joint was approximately equal to 605 MPa; after heat treatment, the tensile strength of the upper, middle, and lower parts was approximately equal to 543 MPa. The tensile strength of the as-welded joint was 10.2% greater than the heat-treated tensile strength.

Figure 4 shows the changes in the section shrinkage rate and the elongation rate of the as-welded and heat-treated joints. From Figure 4a, it can be seen that the elongation rate of the upper, middle, and lower parts of the welded joint was equal to about 6.4%, and the elongation rate of the upper, middle, and lower parts of the heat-treated joint was equal to about 7.4%. The heat-treatment elongation rate was 13.5% greater than the as-welded elongation rate. It can be seen from Figure 4b that the section shrinkage rates of the upper, middle, and lower parts of the as-welded joints were roughly the same, with an average of 68.3%, and the heat-treated section shrinkage rate of the upper, middle, and lower parts were roughly the same, at about 71.7%. The heat-treatment section shrinkage rate was 4.74% greater than that of the as-welded state.

Figure 5 shows the change in the impact energy in the weld center and the fusion line area of the as-welded and heat-treated joints. The standard deviation error bar represents the deviation from the average value, reflecting the fluctuation of toughness. It can be seen that the impact energy of the upper, middle, and lower parts of the weld center of the as-welded joint had a small change of about 141 J; the impact energy of the fusion line area had a large difference: the upper part was about 130 J, the middle part was about 113 J, and the lower part was about 96 J, with an average of 113 J. The toughness of the fusion line was lower than that of the weld center. After heat treatment, the impact energy of the upper, middle, and lower parts of the weld center had a small change of about 183 J, and the impact energy of the upper, middle, and lower parts of the fusion line area had a small change of about 95 J. The toughness of the fusion line was significantly lower than the toughness of the weld center.

The toughness of the heat-treated weld center was 23.0% larger than that of the as-welded seam, and the toughness of the heat-treated fusion line area was 16.0% smaller than that of the as-welded fusion line area. This significant change can be attributed to the microstructure change caused by the heat-treatment-process.

### 3.2. Micro-Hardness Testing

Figure 6 shows the changes in the microhardness of as-welded and heat-treated joints. Figure 6a,b shows that the hardness of the upper, middle, and lower parts of the joint did not change significantly. The average hardness value of the weld center of the as-welded seam was about 18.7% greater than that of the heat-treated joint, and the average hardness value of the as-welded heat-affected zone was about 21.1% higher than that of the heat-treated value. The hardness of the weld state at base metal was equal to that of the heat-treated metal.

In summary, after heat treatment at 580 °C (20 h), the toughness of the weld metal was improved without reducing the strength and hardness of the metal to a large extent.

### 3.3. Microstructure and Fractography Analysis

Figure 7 shows the SEM microstructure analysis of the weld center in the as-welded and heat-treated metal. From Figure 7a,c, it can be seen that after heat treatment, the columnar crystal regions along the grain boundaries of the larger carbides transformed into a large number of fine particles. From Figure 7b,d, it can be seen that after heat treatment, a large number of fine and dispersed granular carbonizations were precipitated at the subgrain boundary in the fine-grained region. In summary, this was mainly because the carbide was dissolved during heating, and then dispersed and precipitated during the cooling process. The fine granular carbides pierced, and hindered the movement of, the dislocations, thereby improving the toughness of the joint after heat- treated. And during the heat- treated, the granular bainite evolved to a softer ferrite matrix. Therefore, after heat- treated, the tensile strength and hardness of the weld decreased, but the toughness increased.

Figure 8 shows the microstructure analysis at the fusion line of the as-welded and heat-treated joints. From Figure 8a, it can be seen that the weld near the fusion line increased the Cr content due to the melting of part of the base material, and a large quantity of rod carbide appeared. According to Figure 8b, the rod carbides were the M_3_C-type carbide. It can be seen from Figure 8c that after heat treatment at 580 °C (20 h), the quantity of rod carbides was greatly reduced, and granular carbides appeared, which reduced the toughness and hardness at the fusion line. Figure 8d shows that the granular carbides were the M_23_C_6_-type carbide [5,18,19]. According to the element distribution in Figure 8b,d, after heat treatment, the Cr content in the carbide increased and the Mn content decreased.

SEM was used to characterize the as-welded and heat-treated tensile fractures. Figure 9 shows the fracture morphology of the as-welded and heat-treated joints. The tensile fractures of the upper, middle, and lower parts of the as-welded joints are shown in Figure 9a–c. The fractures were rugged and formed a large number of dimples, showing ductile fracture characteristics. After heat treatment, the tensile fractures of the upper, middle, and lower parts of the as-welded joints can be seen in Figure 9d–f. There were also a large number of dimples formed on the fractures. The dimples were small and large in number. The ductile fracture characteristics were more obvious. The continuous small surface platform showed the characteristics of a brittle fracture. The tensile fracture mode changed from a ductile fracture to a combination of a brittle fracture and a ductile fracture. Therefore, the tensile strength after heat treatment was lower than that of the as-welded joint, but the elongation and reduction of the area were higher than that of the as-welded joint.

Figure 10 shows the micromorphology of the impact test fracture surface. According to Figure 10a, the left side of the impact fracture in the as-welded joint was the fibrous zone, which was the starting point of the crack. In the middle was the radiation zone, which was the crack propagation zone. On the right was the shear lip zone, which was completely fractured. It can be seen from Figure 10b that the fracture mode after heat treatment was the same as that of the as-welded joint.

SEM was used to characterize the microstructure of the as-welded and heat-treated samples, and to evaluate their fracture modes. Figure 11 shows the fracture morphology of the weld center of the as-welded and heat-treated joints. As seen in Figure 11a, The weld center fibrous zone of the as-welded joint showed a large number of dimples, which characterized a ductile fracture; in Figure 11b, the weld center radiation zone of the as-welded joint showed a large number of cleavage planes, which characterized a brittle fracture; in Figure 11c, the weld center central fibrous zone of the heat-treated joint showed dimples that were smaller and more numerous than those of the as-welded joint. Therefore, the toughness was greater than that of the as-welded joint; in Figure 11d, of the weld center radiation zone of the heat-treated joint, the zone of the cleavage planes were relatively small. In summary, after heat-treated, the fracture mode was converted from a brittle fracture failure mode to a combination of a brittle fracture and a ductile fracture. Therefore, the impact performance of the weld center after heat treatment was better than that of the as-welded joint.

Figure 12 shows the fracture morphology at the weld line between the as-welded and the heat-treated joints. As shown in Figure 12a, the fibrous area in the as-welded fusion line area showed a large number of dimples, and characterized a ductile fracture; in Figure 12b, the radiation area in the as-welded fusion line area showed a large number of cleavage planes, which characterized a brittle fracture; in Figure 12c, the fibrous area of the heat-treated fusion line area shows that the dimple size was equivalent to that of the as-welded joint; in Figure 12d, the radiation area of the heat-treated fusion line area shows that the area of the cleavage planes was relatively large, and the brittle fracture characteristics were more obvious. In summary, the brittle fracture characteristics were more obvious after heat-treated. Therefore, the impact performance of the heat-treated joint fusion line area was slightly reduced.

## 4. Conclusions

The tensile fracture mode changed from a ductile fracture to a combination of a brittle fracture and a ductile fracture. After heat treatment at 580 °C (20 h), the tensile strength of the joint changed from 605 to 543 MPa.

The fibrous zone of the impact fracture was shown to be a ductile fracture. The radiation zone of the weld center was transformed from a brittle fracture to a combination of a brittle fracture and a ductile fracture. The characteristics of the brittle fracture in the radiation zone of the fusion line were more obvious. After heat treatment at 580 °C (20 h), the impact energy at the center of the weld changed from 141 to 183 J. The impact energy at the fusion line changed from 113 to 95 J.

The average hardness value of the weld center of the as-welded seam was about 18.7% larger than that of the heat-treated joint, and the average hardness value of the as-welded heat-affected zone was about 21.1% higher than the value of the heat-treated joint.

After heat treatment at 580 °C (20 h), the matrix structure in the center of the weld was transformed from granular bainite in the as-welded joint to a ductile ferrite matrix. In addition, a large number of dispersed carbides were precipitated in the subgrain boundaries. The carbide precipitation phase of the weld near the fusion line was reduced.

After heat treatment at 580 °C (20 h), the toughness of the weld metal was improved, and the strength and hardness of the metal were not reduced to a large extent.

## Figures and Tables

**Figure 1 materials-14-04498-f001:**
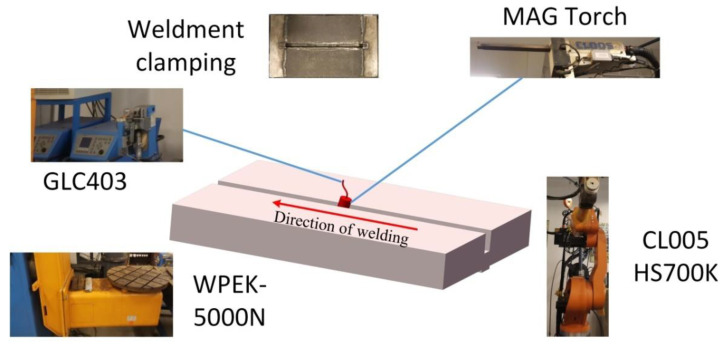
Narrow gap swing arc (MAG) welding system.

**Figure 2 materials-14-04498-f002:**
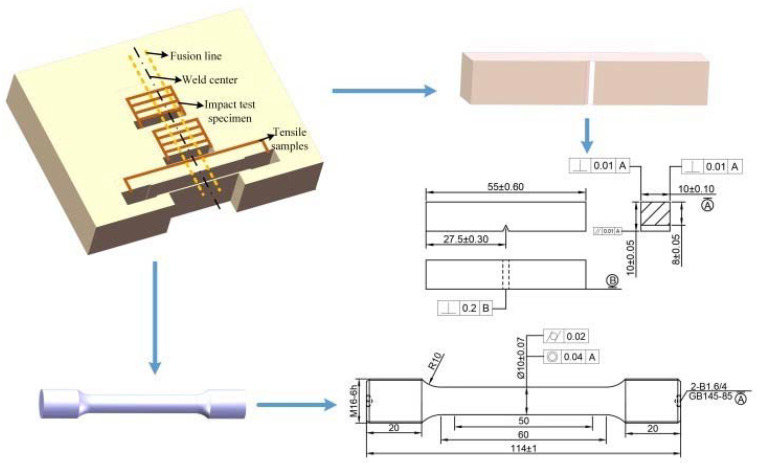
Sampling drawings of tensile and impact specimens.

**Figure 3 materials-14-04498-f003:**
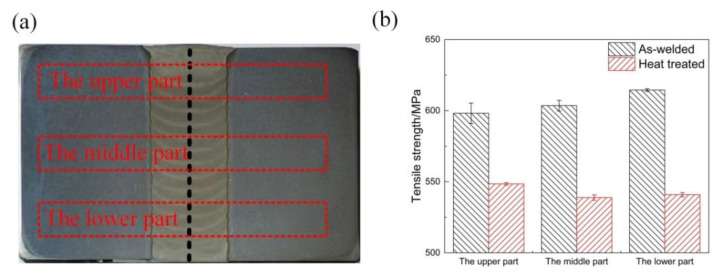
(**a**) Test positions of the upper, middle, and lower parts; (**b**) comparison of tensile strength between the as-welded and heat-treated joints.

**Figure 4 materials-14-04498-f004:**
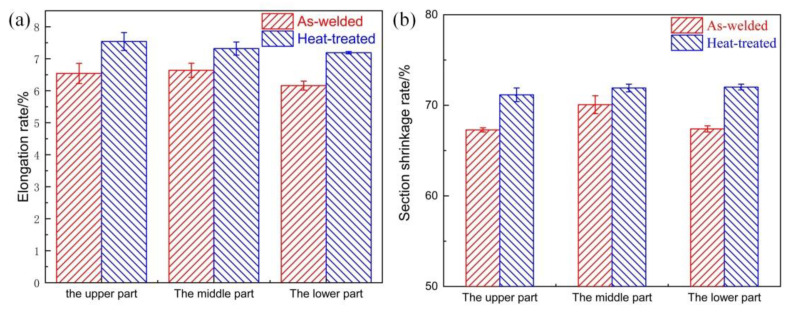
Comparison of the elongation rate and section shrinkage rate between as-welded and heat-treated joints: (**a**) elongation rate; (**b**) section shrinkage rate.

**Figure 5 materials-14-04498-f005:**
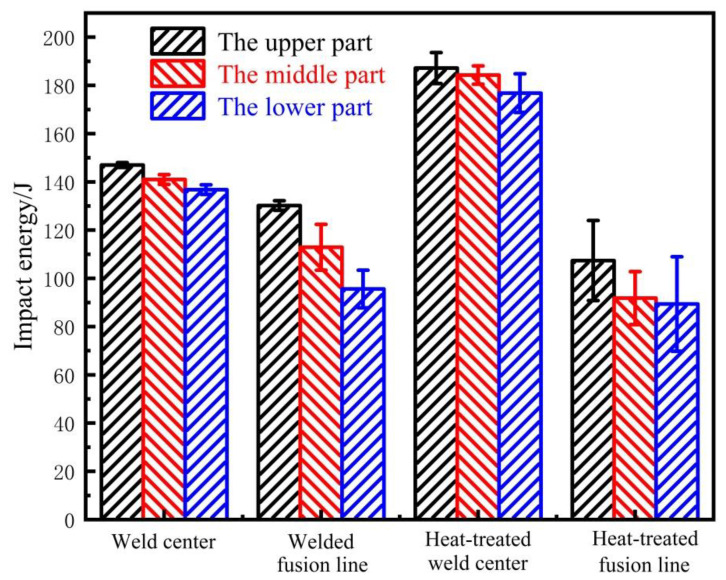
Impact energy in weld center and fusion line area.

**Figure 6 materials-14-04498-f006:**
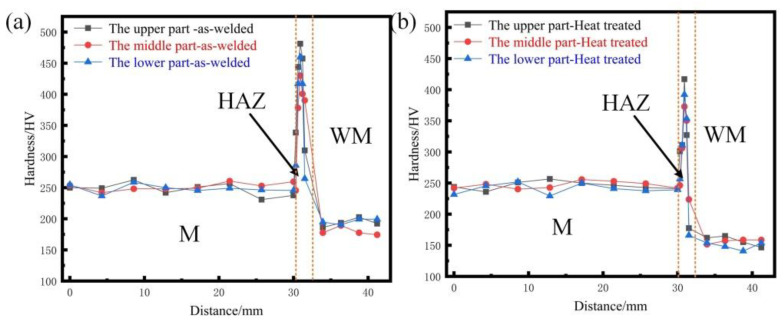
Microhardness results: (**a**) microhardness test position; (**b**) microhardness comparison of the as-welded joint; (**c**) microhardness comparison after heat treatment.

**Figure 7 materials-14-04498-f007:**
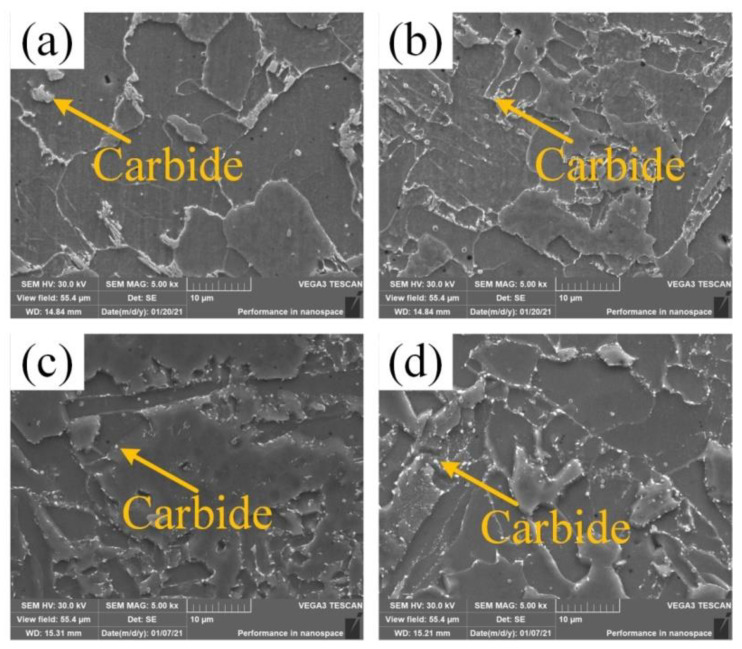
SEM was used to analyze the weld center structure: (**a**) as-welded state coarse grain zone; (**b**) as-welded fine grain zone; (**c**) heat-treated coarse grain zone; (**d**) heat-treated fine grain area.

**Figure 8 materials-14-04498-f008:**
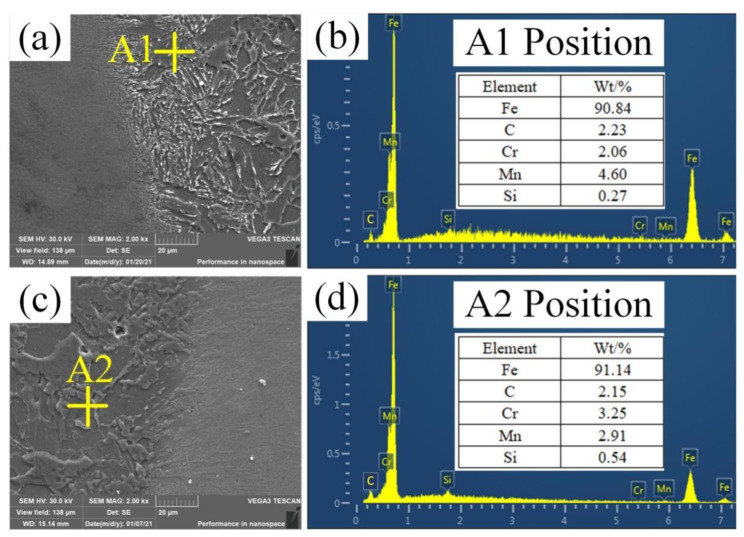
Microstructure analysis at the fusion line: (**a**) SEM images of the as-welded joint; (**b**) the EDS result of point A1 in (**a**); (**c**) SEM images of the as-welded joint; (**d**) the EDS result of point A2 in (**c**).

**Figure 9 materials-14-04498-f009:**
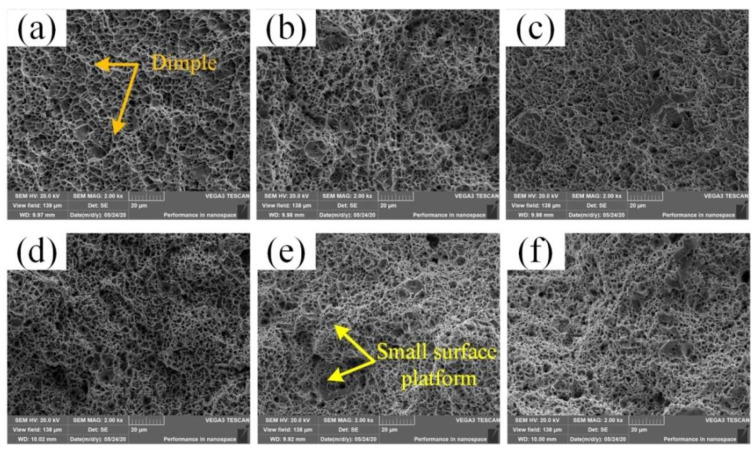
(**a**–**c**) Fracture of the tensile specimen at the upper, middle, and lower parts of the as-welded joint; (**d**–**f**) fracture of the tensile specimen at the upper, middle, and lower parts of the heat-treated joint.

**Figure 10 materials-14-04498-f010:**
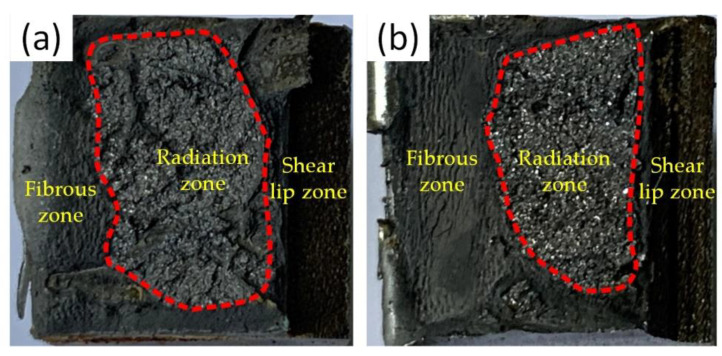
Micromorphology of impact test fracture surface: (**a**) as-welded; (**b**) heat-treated.

**Figure 11 materials-14-04498-f011:**
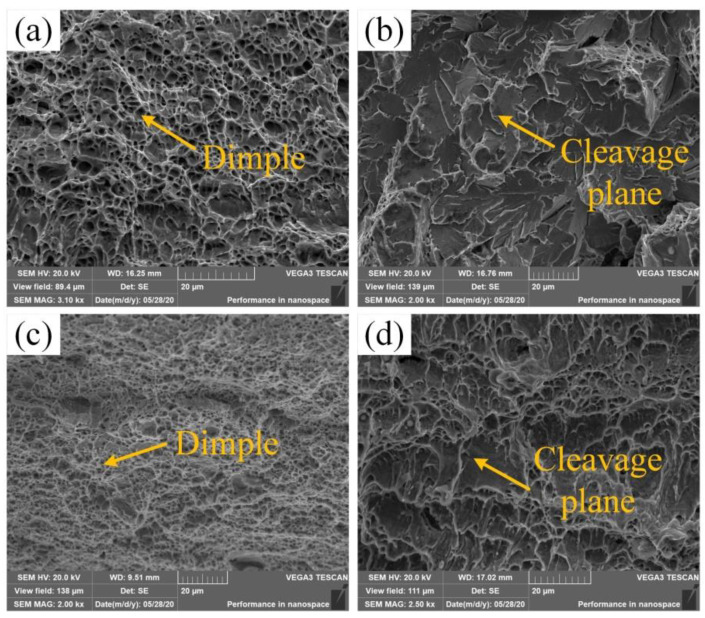
Fracture morphology of the weld center joint: (**a**) the weld center fibrous zone of the as-welded joint; (**b**) the weld center radiation zone of the as-welded joint; (**c**) the weld center fibrous zone of the heat-treated joint; (**d**) the weld center radiation zone of the heat-treated weld.

**Figure 12 materials-14-04498-f012:**
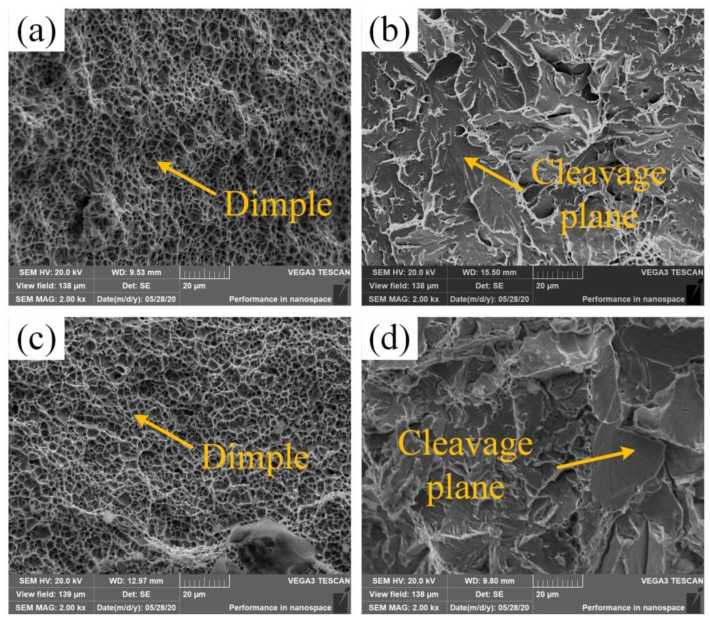
Fracture morphology of the fusion line area: (**a**) the fusion line fibrous zone of the as-welded joint; (**b**) the fusion line radiation zone of the as-welded joint; (**c**) the fusion line fibrous zone of the heat-treated joint; (**d**) the fusion line radiation zone of the heat-treated joint.

**Table 1 materials-14-04498-t001:** 25Cr2NiMo1V base metal and JM-56 welding wire alloy content (wt.%).

	C	Si	Mn	Mo	Cr	V	P	S	Cu	Ni
Metal	0.26	0.31	0.77	0.95	2.4	0.41	0.031	0.028	0.23	0.27
JM-56	0.077	0.87	1.45	0.002	0.031	0.004	0.012	0.013	0.125	0.017

## Data Availability

Date sharing not applicable.

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
