# Peer review of "Effect of 580 °C (20 h) Heat Treatment on Mechanical Properties of 25Cr2NiMo1V Rotor-Welded Joints of Oscillating Arc (MAG) Narrow Gap Thick Steel"

_materials, 2021, doi:10.3390/ma14164498_

Round 1

Reviewer 1 Report

First of all, the reviewer would like to urge the authors to carefully revise the paper grammatically, formatically and linguistically. The readability of the paper must be significantly improved. More importantly, the scientific merits of the paper is weak. The authors did characterise the fusion boudary but the scientific story behind is not fully comprehensive clarified, e.g. what is the effect of heat treatment on the fusion boundary microstructure (boundaries evolution, elemental diffusion, the range of HAZ, etc.)? Which kind of carbide was precipatated at the fusion line. What is the influence of carbides evolution on the cross-fusion boundary micro or nano-hardness?

Reviewer 2 Report

The manuscript under review analyzes the effect of 580℃ (20h) heat treatment on mechanical properties of 2 25Cr2NiMo1V rotor-welded joints of oscillating arc (MAG) 3 narrow gap thick steel. An extended experimental analysis is provided by the authors however besides the results the presentation fails to analyze them deeply and to correlate microsrtuctural features to the mechanical properties of the welds.

Some specific comments

Page 3, lines 79-83: The word "about" is mentioned several times. A more accurate estimate is needed in all cases.

Page 3 line 91: "heat treatment process of 580℃ (20h)" How did the authors selected  this temperature?  In addition how did choose the time of 20h ? Are there any calculations about this issue?

Page 4, line 110: "styles" The expression needs rephrasing.

Page 4, lines 109-110: Rephrasing is needed

Page 4, Fig. 3: Upper Middle and lower: The authors must specify what this characterization means.

Page 5, lines 119-120: "The tensile  strength of the welded joint is 10.2% greater than the heat-treated tensile strength." Why this happened? An explanation must be provided by the authors

Page 6, Figure 1: This is not Figure 1 must be Figure 6

Page 7, line 175:being ab-175 "absorbed during heating"  Dissolved is the proper expression here

Page 7, line 177 "Rod carbide is M3C type carbide, and granular carbide is M23C6 type   carbide"  It is not clear from the provided SEM images the type of carbides. An EDS analysis should be provided

Page 8, line 199:Figure 2: This must be Figure 9

Reviewer 3 Report

The paper titled "Effect of 580°C (20h) heat treatment on mechanical properties of 2 25Cr2NiMo1V rotor-welded joints of oscillating arc (MAG) narrow gap thick steel" provides a description of the characterization of welded joints in 25Cr2NiMo1V based on tensile and impact tests. Despite the topic is of interest, the paper must undergo major revisions, starting from a global check to the manuscript format (some captions are not ordered and sometimes the manuscript refers to the wrong picture). Beside the general recommendation, the authors are asked to address the following list: 1. The combination of temperature and time for the post welding heat treatment (580°C/20 h) is taken from literature? if so, the paper should be cited. 2. The description of Figure 2 has to be improved, since the extraction scheme is not clear enough (or, at least, it is not sufficiently detailed). The tensile specimens are extracted so that the weld line was perfectly centred over the gauge length. For the impact tests, it seems that two different types of specimens were extracted: a first one, for which the weld line was in correspondence of the v-notch and the other for which the fusion line was aligned with the v-notch. Is it true? If so, the extraction strategy has to be better detailed. 3. Lines 109-111 are not easily readable: what is the meaning of "pattern" and "styles"? 4. Since the error bands in Figure 3 are not easily visible (due to a small deviation standard among the replications), it would be better if the deviation is mentioned in the text (at least for the results coming from the tensile tests). 5. Line 130: Figure 8b does not refer to the reduction of the cross section (which is more indicated than "shrinkage"). 6. In some parts, the paper gets too confusing, which is mainly due to the lack of coherency in the nomenclature: at first the authors should refer to the two parts from the welded butt joints as "post-welded" and "heat-treated" and keep such a nomenclature throughout the whole manuscript (sometimes it is called welded, other times post-welded, then as-welded). The same consideration yields also when the three zones of investigation are labelled at first as "Upper, Middle and Lower" and then "Top, Middle and Bottom". The nomenclature should be always coherent. 7. Figure 6a (even though is referred to a wrong "Figure 1" caption on line 154) should be moved earlier in the manuscript: the definition of the three zones of investigation should help the comprehension of the reader from the beginning, i.e. from the results of the tensile tests. Moreover, the hardness distribution in Figure 6b and Figure 6c refer to a 0 position which is neither discussed in the manuscript nor indicated in Figure 6a. 8. Line 173-179 Among the 4 details in Figure 7, it is not clear which are the ones referring to the weld state (assuming that "weld state" has the same meaning of "post-welded" state) and those to the heat-treated condition. 9. From line 220 on, the authors mention the "central fiber area" and the "central radiation area" that are not clearly distinguishable in the pictures and are not sufficiently detailed. 10. As a general consideration, there is a lack of discussion about how the carried out methodology can provide a guideline for the investigated welding technique.

Round 2

Reviewer 1 Report

Thank the authors for the revisions!

Author Response

Response: Thanks for your comment. It has been modified.

Reviewer 3 Report

The comments provided by the Reviewer have been satisfactorily addressed by the authors and the paper is suitable to be published.

Author Response

(The authors gave the same response as above.)
